# Leveraging Faith Communities to Prevent Violence against Women: Lessons from the Implementation and Delivery of the Motivating Action through Empowerment (MATE) Program

**DOI:** 10.3390/ijerph192315833

**Published:** 2022-11-28

**Authors:** Karen Pearce, Erika Borkoles, Sharyn Rundle-Thiele

**Affiliations:** 1School of Medicine and Dentistry, Griffith University, Southport, QLD 4222, Australia; 2Social Marketing @ Griffith, Griffith University, Nathan, QLD 4111, Australia

**Keywords:** violence against women, domestic violence, violence prevention, bystander approach, train-the-trainer, faith community, community intervention, COM-B model

## Abstract

Gender-based violence is a human rights and public health issue, disproportionately affecting women. The Motivating Action Through Empowerment (MATE) bystander program aims to address violence against women by shifting focus from perpetrators and victims of violence to community responsibility for not accepting attitudes and behaviors that support or allow the violence to occur. Traditionally bystander programs have been delivered through institutions, most notably college campuses in the United States. The translation of bystander programs to community settings is not widely reported. This research aimed to understand whether a violence prevention program could be effectively delivered in a faith community setting; specifically, it focuses on the implementation of MATE in a Christian church network in the Gold Coast region of Queensland, Australia. Semi-structured interviews were conducted with ten church-based trainers in the MATE pilot program. Theoretically informed analysis using the COM-B behavior model identified that environmental factors had a large bearing on opportunities to deliver MATE workshops. This research identified six key lessons for MATE and other programs wishing to leverage faith communities: (1) Provide religious context; (2) Accommodate diversity; (3) Build faith leader capacity; (4) Employ social marketing; (5) Undertake co-design; (6) Actively administer, measure and monitor.

## 1. Introduction

Gender-based violence is a global human rights and public health issue that disproportionately affects women. In Australia, 1 in 5 women have experienced sexual violence from age 15, compared to 1 in 20 men; 1 in 6 women, aged 15 and over, have experienced stalking, compared to 1 in 15 men; and women are twice as likely as men to have been sexually harassed since the age of 15 [1]. In 2020, 84% of the 27,505 victims of sexual assault recorded by police nationally were female. Nearly 1 in 5 of these assaults occurred in Queensland, where victims were six times more likely to be female than male [2].

There are clear predictors of rates of violence against women. These are condoning violence against women; men’s control over decision making and limiting women’s independence in private and public; rigid and stereotyped gender roles; and male peer relations that support inappropriate attitudes and behavior towards women [3]. Circumstances favoring these predictors are enhanced in times of uncertainty, crisis and disaster, such as floods, fires and pandemics [4]. The increase in domestic violence after natural disasters in Australia has been documented [5], as has the increase in the severity and prevalence of domestic violence due to COVID lockdowns [6]. With climate change likely to exacerbate natural disasters [7] and uncertainty about ongoing implications of the COVID pandemic, violence against women will continue to be a major public health issue.

One form of intervention to reduce domestic violence is a bystander program. Bystander programs aim to change social norms about violence against women [8]. They were first used in the United States in the 1990s with the introduction of the Mentors in Violence Prevention (MVP) program [9]. This program, established at Northeastern University in Boston, engaged college athletes to drive cultural change, recognizing their position as masculine role models. The athletes were supported to demonstrate and promote positive and respectful attitudes and behaviors towards women with the intention of inspiring their peers to do likewise [9]. Since the successful implementation of the MVP program, a range of bystander programs have been developed, predominantly for delivery in university settings [10,11]. More recently, these programs have been modified and delivered in high schools [12,13]. Bystander programs have also been trialed in the military [14] and in workplaces [15,16]. Translation into community settings has not been widely reported. One notable exception is the Active Bystander Communities program in the United Kingdom [17]. In this program, the absence of strategic interest, captive audience, shared identity and supported delivery spaces common in university populations (and conceivably in workplaces, the miliary and any institutional setting) posed challenges for translating bystander programming to a general community setting [17]. General community settings also lack access to the necessary funding and resources to sustainably run such a program [18].

Another notable omission is engaging faith communities in bystander programs. Faith communities are groups of people who share beliefs and practices based in religion or spirituality. The structured nature of religious faith communities presents many opportunities and resources for health promotion, including venues, motivated people, a range of skills and established communication networks [19]. Faith communities could offer an ‘intermediate’ community setting with regard to community-based violence prevention programs, with more structure and resourcing in place than a general community setting but far less than institutional settings. For this reason, faith communities may offer a useful setting for translating bystander programs for use in the broader community. Additionally, faith communities have been identified as priority settings for action to address violence against women because of their influence on social norms and beliefs and their delivery of community programs and services [3]. However, empirical evidence regarding the success of violence prevention initiatives in faith settings is limited [20].

This research examines the Motivating Action Through Empowerment (MATE) bystander program. It is based on the MVP program, which was introduced in Australia in 2010 through Griffith University’s Violence Research and Prevention Program. The program was later modified for the Australian context and launched as MATE in 2016 [21]. The program’s primary intervention is a cultural change workshop that aims to address violence against women by making people aware of violence’s roots in gender inequality and providing participants with a framework for taking prosocial bystander action. MATE began delivering the workshop in 2016, predominantly to corporate organizations on a fee-for-service basis. In 2017, a train-the-trainer course was added so that organizations could deliver their own cultural change workshops in-house, easing the resource burden on the small MATE team. This delivery model, which has seen the program delivered across Australia, has been successful for MATE [22]; however, the focus on corporates does not accommodate participation by grassroots community members.

This research evaluated a MATE pilot program that aimed to determine if church-based trainers can serve as an effective means of delivering their violence prevention program to communities. The pilot, in the Gold Coast region of Queensland, Australia, was conducted in partnership with a not-for-profit organization (NFP) facilitating coordination and collaboration between Christian churches on the Gold Coast with a view to enhancing their social impact. In the pilot model, the NFP recruited and facilitated Christian church involvement in the MATE program. Participating churches pay a fee for members of their communities to undertake a MATE train-the-trainer workshop with participants from other Christian churches recruited by the NFP. Participants may be faith leaders or congregants. On completion of the workshop, participants are accredited and encouraged to deliver the MATE program thorough their church ministries. After 12 months, trainers are required to undertake a refresher workshop to maintain their accreditation. The licensing arrangement underpinning this model does not allow the trainers to deliver MATE content beyond their church ministry. In this way, church community members can access MATE workshops at no cost, while MATE maintains the ability to generate income from the program through other activities.

This study seeks to contribute knowledge about how the program is being adopted and delivered in this faith community setting. The COM-B model was used to identify barriers and enablers of the program’s implementation. This model, which posits that a behavior (B) is the result of the interplay between capability (C), opportunity (O) and motivation (M) [23], provides a relatively simple and easily applied guide to understanding influences on behavior that readily encompasses personal, inter-personal and environmental factors.

## 2. Materials and Methods

### 2.1. Program Implementation

Program implementation consisted of delivery of the three-day MATE train-the-trainer workshop to the church-based participants recruited by the NFP partner. The program was not modified from the standard version offered to MATE participants outside this pilot. The workshop was delivered by MATE facilitators and covered leadership, bystander intervention, the nature of violence against women, and gender inequality. On completion of the workshop, trainers were given the standard MATE trainer manual and PowerPoint slide deck to use in their program delivery. Trainers were invited to join an online Community of Practice on Microsoft Teams and to join a private MATE Leadership group on Facebook. For the pilot cohort, an additional private Facebook group was also established. Following the workshop, trainers received topic- and program-related communications from MATE and the NFP on an ad hoc basis by email and through the online groups.

### 2.2. Study Design

This study used a collective case study design [24] to gain insights into the implementation and delivery of MATE in the participating church communities. This design was selected because it was important to understand the implementation of MATE in this specific context, as the setting is the novel component of this delivery model. As some church-based trainers had already delivered MATE workshops, this study also offered insights on visible impacts of the program in participating church communities.

### 2.3. Study Participants

Study participants were drawn from the list of 31 individuals from 12 Gold Coast Christian churches (Table 1) who participated in the MATE pilot train-the-trainer workshops conducted in May 2021 (n = 16) and July 2021 (n = 15).

A non-probabilistic convenience sampling approach was used, which relied on trainers volunteering to take part. Recruitment was via direct invitation from MATE and the NFP partner. All church-based trainers who participated in the two train-the-trainer workshops, and who did not hold previous MATE training accreditation, were eligible for inclusion in the study. The study sample size was n = 10, comprised of representatives from seven churches. Study participants were characterized by age, gender, average weekly household income, highest level of education, and cultural heritage (Table 2).

### 2.4. Data Collection

Trainers who indicated their willingness to participate were contacted by the researcher to set an individual interview time. Participants were emailed a link to an online trainer registration survey for completion prior to the interview. This survey collected demographic information (age, gender, average weekly household income, highest level of education, and cultural heritage) as well as background information about their involvement in the MATE program (how they heard about the program, why they got involved, which of the two train-the-trainer workshops they participated in, how many workshops they had delivered, their intentions for delivering MATE in the future, and whether they participated in the online MATE community of practice).

Semi-structured interviews were conducted online in November 2021 and January and February 2022 using Microsoft Teams. The first author (KP) conducted all interviews using an interview guide. Interviews lasted between 21 and 46 min (mean = 36.4; SD = 8.35) and were recorded with the interviewee’s consent using Teams. The interview guide was tested with a church-based trainer who participated in the pilot but had previous experience with the MATE program, so was excluded from the study. No changes were made to the guide following this interview. Interviews were transcribed.

### 2.5. Data Analysis

Interview data was inductively thematically analyzed [25,26] to identify primary emergent themes relating to how the MATE program was introduced to and set up in the community (implementation) and to understand what the delivery of the MATE workshops looked like (delivery). Themes were refined by a review of the codes to remove redundancy and ensure common ideas were coded similarly. To determine what influenced delivery of MATE workshops (delivery influences), themes were deductively mapped to the six sub-components of the COM-B behavior model: capability (psychological and physical), opportunity (physical and social) and motivation (reflective and automatic) [23].

## 3. Results

### 3.1. Implementation

The primary implementation activity in the MATE pilot is the train-the-trainer workshop. For trainers with some experience of violence against women, the workshop provided a vocabulary and theoretical context around the issues they were already familiar with. For others, the workshop introduced the extent and nature of violence against women, providing many with new information and perspectives. The workshop also introduced the MATE program and how it is delivered. Opportunities to consider and respond to issues that may be encountered when delivering the MATE program were highly valued by the church-based trainers.


*Yeah, oh look, it was a very… it was an eye opener, you know, to get an understanding from the perspective that they [the MATE trainers] were coming from and just getting all the stats, I suppose, of what has been happening out there in the public.*
(T06)


*That was very useful to be able to go OK, how to deal with the different types of people that you’ll come up against, whether it be positive or negative.*
(T06)

The training manual provided to each trainer at the conclusion of the workshop was also valued by the church-based trainers. They were generally confident that with this resource, which included both course content and guidance on how to deliver it, they would be able to run their own MATE workshops. However, perspectives varied as to how strictly the manual needed to be followed.


*I think the booklet that we went away with was the most useful, practically, to then go and deliver, you know, kind of be able to facilitate something for your community.*
(T09)


*[I]f you do the MATE train-the-trainer course … we don’t get to pick and choose the topics that are delivered to us… you have to look at every single part of that wheel. Whereas when you’re delivering it and you can deliver it how you want, when you want, for whatever, there are some things that just get mixed up, like missed out.*
(T04)

Trainers were given access to two online support communities: a community of practice hosted on Microsoft Teams for all accredited MATE trainers (i.e., not just those in this pilot) and a private Facebook group just for the pilot cohort. None of the interviewed trainers had participated in the all-trainer community, and those that used the private Facebook group has mixed experiences. There was also some confusion over the purpose of the different online communities. In general, however, trainers felt that support from MATE and the NFP partner was available to them should they need it.


*I think there have been posts and meetings and stuff that could, you know, we could have attended […] but we haven’t been able to access the content of that.*
(T05)


*[O]ne thing I was disappointed with was we had a Facebook page that started up from the training, like for our group […] But there’s really been very little interaction on that.*
(T07)


*So, we do have like the MATE alumni group and stuff. And for me, I’m not entirely sure what the crossover or lack thereof is between [the NFP partner] and the MATE stuff I’m connected with online. I’ve seen it more as like a story sharing space rather than a support space in terms of like, I probably wouldn’t go there and be like, hey, I’m really struggling with this aspect of it all.*
(T03)

A common area of concern identified by trainers was the lack of religious or biblical perspective presented with the MATE content. Given the Christian context of this pilot, offering a religious perspective was considered an important component of the MATE program. This perspective was necessary to both endorse the MATE program and to counter ‘faith-based’ opposition to the ideas and issues raised in the program. Trainers expressed some frustration that this perspective was not fully addressed in the MATE train-the-trainer program.


*[O]ne thing that was… hadn’t been formulated at that point was the biblical perspectives or, you know, aligning, I guess, of those worlds. And that was definitely something that we were waiting on prior to starting any of our training. Because we knew that the church world, Christians, were going to want to have that perspective in there.*
(T05)

### 3.2. Delivery

At the time of interview, four of the 10 trainers reported that they had delivered two MATE workshops, four had delivered one workshop and two had not delivered a workshop. All trainers indicated their intention to deliver MATE workshops in the future.

Church leadership and staff was a priority audience in three of the seven participating churches, and of the 10 MATE workshops that had been delivered at the time of interview, half had been delivered to staff (Table 3). Reasons for this included getting leadership buy-in and ensuring staff were aware of MATE content before it was delivered to the broader congregation. The format and content of the workshops varied between churches.

### 3.3. Delivery Influences

Twenty-seven influences contributing to trainers’ delivery of the MATE program were identified in the interviews and categorized into 12 themes. These themes were aligned to the trainers’ capability, opportunity or motivation to deliver the MATE program following the COM-B behavior model [23] (Table 4).

#### 3.3.1. Capability Influences

The MATE train-the-trainer workshop was successful in opening trainers’ eyes to the extent of violence and extending their understanding of the drivers of violence against women. Knowledge about the bystander intervention framework, the MATE program’s guide for taking prosocial bystander action in the face of violence against women, was only mentioned by one trainer, and that person could not recall the framework in its entirety. All trainers acknowledged the large volume of content in the program; however, they felt that they could rely on the resources they had been provided with to assist their own program delivery. Trainers had a range of facilitation experience, and many expressed that they were at ease with having to get up in front of people to present. However, one trainer (with facilitation experience) believed that more practice in this area was required to develop the requisite skills.


*I really had no background knowledge on the whole gender side of things and how that influences, you know, the process.*
(T06)


*I was thinking abuse, so I was thinking physical, yeah… and I wasn’t thinking everything else, you know, that we were taught and discussed.*
(T10)


*[Y]ou know, distraction, protocol, intervention or… I can’t remember the fourth one right now.*
(T05)


*[T]he booklet I think was amazing because it had all the topics, you know, and then just detail around how you could facilitate that.*
(T09)


*[M]y concern is around how then do people take what they’ve covered in training, in a two-day period, and do that then well with their communities.*
(T09)

#### 3.3.2. Opportunity Influences

Church environment: Church leadership emerged as a critical element of the church environment, as the leadership ultimately makes the decisions about implementing and maintaining the MATE program within their ministry. Churches with strong leadership support have a clearer vision of how MATE will fit into their ministry, and in many instances have taken steps to make that happen. In churches where leadership support is not as clear, MATE has remained on the periphery.


*Well at this stage I would say that we are kind of on the fringe… we haven’t actually got it as being sort of accepted as a formal ministry yet… I think [the church leadership are] supportive of doing the, or the idea of having [MATE] happen. I think they, certainly some of them, feel they’re not sure how it’s gonna fit. And for some of them, they kind of go, well, it’s probably, you know, they have that kind of denial—well, is this something that we actually have in our church?*
(T02)

In general, the trainers expressed a strong sense of the difficulty of the MATE content for a faith setting, with many suggesting omissions and modifications that would be made. The topic of gang rape, in particular, was perceived by trainers as being confronting and/or less relevant to the church audience. Others, however, acknowledged the difficult content as something that needed to be faced by the congregation. One trainer voiced their belief that Christians would not come across some of the behaviors covered in the MATE program, suggesting limits to acknowledgement of violence against women and precursory behaviors in the church.


*You know, like delivering [the gang rape topic] in the church context is just, you know, like that just, that will just blow people’s brains, you know. That’s just not something that most people following a Christian way of life would even encounter.*
(T07)


*[W]e’ve kind of had to be selective in what subjects you kind of choose… I think some content would be more shocking than others.*
(T05)

Scheduling is a big issue for getting MATE into church ministry, both in terms of fitting it into already full calendars and with regard to the time the delivery team has available to deliver a MATE program. Some churches have decided to leverage these existing ministry activities events, such as men’s program meetings and youth groups, and use them as opportunities to deliver MATE content instead of scheduling standalone MATE workshops.


*[I]t’s been fully supported. We think, yeah, we think it’s awesome…it’s just a timing thing now. Just to, trying to get it into the schedule.*
(T08)


*[W]e realized that we need to start somewhere, and the men are willing to do it, and they’re already meeting once a fortnight anyway, so we’re just going to infiltrate a couple of their nights.*
(T08)

Resourcing and logistics: COVID-related issues had an impact on the delivery of MATE in this pilot. Apart from lockdowns, which prevented people travelling and meeting, COVID affected the availability of resources for churches to offer the MATE program. These issues are on top of difficulties due to other time commitments that the trainers have, often stemming from work inside and outside of church. There was a hope that expanding the MATE delivery team in their church would address this. Despite the personal toll of the time required to deliver the program, some trainers were keen to take MATE beyond their church community, acknowledging that not everyone in the broader community attends church. As such, some trainers felt limited by the licensing conditions for the pilot program, which do not permit delivery of MATE beyond the church ministry.


*My biggest problem is I’ve moved into a pretty seriously time draining job at the moment.*
(T02)


*That was the challenge for us, just recognizing that it’s not our, it’s not part of our job description to be teaching it… it’s taking us away from our, from what we actually have to do in our job.*
(T10)


*I think for me, like, if we had a bigger team at church and we had again just more people that were officially on board with it and not just on board, but officially a pilot team on board working towards it, I think that’d be a huge asset moving forward.*
(T03)


*Is it enough just to be in the churches? No, it needs to, we need to develop this so it can be out there in the general community… but it’s then how we get it out into the general public. Because there’s still a lot of people that don’t go to church.*
(T06)

Christian context: Trainers have a strong Christian identity based on being more Christ-like and using guidance from the Bible to achieve that. Related to this, trainers expressed a clear sense of the Christian community being distinct from the non-Christian community. This is due to the central, elevated and explicit role of biblical teaching and guidance in a Christian context compared with non-Christian contexts. Even though they generally thought that the MATE program successfully found common ground, there was a strong belief that the program needed to be shaped differently for Christian and non-Christian contexts. Trainers identified a need for guidance on how to offer biblical insights that support MATE themes and counter contrary Bible interpretations. This guidance was generally viewed as needing to come from members of the Christian community rather than the MATE program itself. The diversity of views and opinions within the Christian environment and within each church community was acknowledged by the trainers, as was the complexity that this could bring to the reception of MATE content.


*[I]n a church environment, we’re there because we believe in the gospel… the story of the gospel, it changes us and it and it calls us to be better people and to make a difference in our community and to become more like Christ.*
(T10)


*[W]e were obviously doing this from, you know, as the Gold Coast churches kind of coming into it, so it was it was good to have, like, [the MATE facilitators] that were doing it, that weren’t coming from that church environment, so they were able to, you know, present it, but also for both of our sides, I suppose to be able to come together and be able to understand.*
(T07)


*I think I was fully prepared to [deliver MATE] in a non-Christian environment. I… it fell short for the Christian environment.*
(T04)

Gender and power dynamics: While the churches support the idea of gender equality, it does not always get applied in practice. This is particularly apparent among church leadership, where a number of trainers remarked on the underrepresentation of women. Conservative views and bible interpretations are challenging for trainers, as they are counter to MATE content. Church hierarchy and power structures also have an impact on MATE delivery as they interfere with participants’ comfort to express views and thoughts freely.


*I would say that we, our church, probably has, like a lot of churches, probably majority women, probably at least 55/45 women to men. And yet a lot of the leaders are probably men.*
(T02)


*But there’s been processes that have happened within our church, and I believe it happens in all workplaces, where that inequity comes from. So, and that’s been a little icy to have that conversation because we have ended up with mostly male leaders in our church.*
(T07)


*Well, when they talk about gender roles… they just flat out won’t go outside that. This is what the Bible says a man is and this is what the Bible says a woman is. So, as a result, they wouldn’t include LGBTI community, like because that falls outside the realm of what they believe the Bible says.*
(T04)


*And then sometimes in those environments you’re just not going to go against your leadership. Whether or not it’s a male or female, you know, like there’s just a whole different dynamic there. I guess it’s like going to a course with your boss and then trying to tell the facilitator that your boss is just flat out wrong, it’s just not going to happen. So that dynamic is there.*
(T04)

Support and encouragement: The trainers felt support from MATE and the NFP partner was available should they need it; however, few have actively sought support. Trainers identified peers from the MATE program and members of church communities as additional sources of support, even though many had not accessed these supports. One trainer who did reach out for support was disappointed with the outcome. They were keen to connect with the peer community, prepare some MATE-related materials, and get some practical advice about delivering to a youth audience. Their expectations of the online community of peer support and practical support from the MATE program were largely unmet.


*[O]ne of the things that I had said when we did our first delivery is we’d like to create a little booklet… [MATE] said, oh, that was a great idea, that was a great idea. Let us work with you. And then, like, I was, like, hey, I’m ready. But yeah, nothing ever, sort of came of that.*
(T07)


*[I] definitely would love to get some more support around, yeah, how we can, some ideas around how we deliver this to, like to youth... rather than, you know, I don’t want to be reinventing the wheel.*
(T07)

#### 3.3.3. Motivation Influences

A major motivator for the trainers stems from their identity as Christians and their belief that they should model their lives on Jesus Christ or do what God wants them to do. There is also a strong sense of responsibility and leadership, both with regard to the role the church should be playing in addressing violence against women and the personal responsibility of the trainers. The trainers were also motivated by the sense of being able to take action and make change happen. They felt very empowered to facilitate positive change through the MATE program, with many echoing the idea that the program showed how everyone’s contribution could lead to greater social change. Trainers also expressed their ‘passion’ about doing something to address violence against women, suggesting that MATE gave them an avenue to follow this passion.


*[W]hen I think about Christ and who he was, and I read about him, he actually shows some amazing points and amazing behavior that I can model after, try and model from as a male in my community, my family.*
(T01)


*[A]s a follower of Christ I believe that one of the greatest things that he did was that he interacted with people where they are at… And so, from my point of view, that’s kind of where I’m at, is that I need to find ways to connect with people. … I’ve gotta be able to find a way to be able to be in their shoes or understand the space they’re in, or be able to, you know, be comfortable in that space so they don’t think that I’m just there to try and fix them up. But I’m actually there to be, you know, like them and to be supportive.*
(T02)


*I feel like this is something that God has really called me to do.*
(T07)


*I feel like it’s just something that, as the church, we’ve got to step up and really make a change and a difference here.*
(T07)


*They actually made me feel like, empowered me to be like, you can actually make a difference, no matter how small or big.*
(T01)


*Like just, [the train-the-trainer workshop] just sparked the passion that I have like inside of me, even more.*
(T07)

## 4. Discussion

A number of areas for improvement emerged in the trainer interviews that cut across program implementation, workshop delivery and influences on workshop delivery. Insights gained from this evaluation can guide how the MATE program is delivered and implemented in faith communities, and inform other programs wishing to leverage faith communities.

### 4.1. Provide Religious Context

Sociocultural relevance of a program to its target community is a recognized success factor in the delivery of community public health and prevention programs [27,28], and in a faith setting including spiritual perspectives is necessary to achieve success [29,30]. In a faith community, alignment of MATE content to biblical content is critical, not only to encourage uptake of messaging but also to head off misinterpretation of Bible content that could be used to block MATE ideas, understanding that teachings and sacred texts can be used by faith leaders to condone violence against women [20].

### 4.2. Accommodate Diversity

While all churches in the pilot have the same core Christian values, there are variations—even within the small sample in this study—in how those values translate to practice, as well as congregational cultural diversity, and differences in staff numbers, resources and congregation sizes. All of these factors make each church—and its requirements from the MATE program—unique. Intra-denominational differences in how churches and church community members practice their faith are complexities to be considered in health promotion efforts in faith communities [31]. The differences in this pilot are reflected in the wide and varied delivery modes that have been used within the MATE pilot program (Table 3).

While MATE presently offers a great deal of flexibility in how the program is delivered, this may be at the cost of losing core content or intentions when modifications are left up to each individual trainer. There is opportunity for MATE to facilitate adaptation of the program to ensure an appropriate fit with each church community and their needs while maintaining program fidelity. This could be achieved in a number of ways, such developing clearer guidelines for how the MATE program can be customized; preparing more ministry-ready MATE guidelines, materials and agendas that can be easily slipped into existing activities; and providing guidance and support on how to embed MATE messages into other ministry activities, such as youth and men’s groups, and in sermons and other regular church activities. As well as addressing barriers of scheduling and resources, this could be a way that churches are able to lead by example and both model MATE in their communities and normalize MATE messaging.

### 4.3. Engage Leadership and Build Leadership Capability

A number of influences on MATE training delivery stem from the structure and operation of each church. These include patriarchal traditions, hierarchies within the church, conservative views, being seen to take prosocial action as a church and leadership support. With the central role they play in operational, attitudinal and behavioral elements of their church communities, faith leaders are in a position to act on all of these influences in a way to facilitate MATE delivery. Faith leaders are influential figures in faith settings and warrant particular attention with regard to implementation of public health and prevention programs [32,33]. This was demonstrated in the trainer interviews, where it was apparent that churches with engaged and supportive leadership were further down the path of delivering MATE in their ministries than churches where this was not the case. A violence prevention program being run through an Australian Anglican church has recognized the importance of engaged leadership and targeted capability building of faith leaders as a core aim of the program to facilitate program outcomes [34]. This approach may have benefit for MATE.

### 4.4. Employ Social Marketing

The current MATE implementation views MATE as a standalone activity. Trainers are expected to deliver MATE workshops using agendas provided in the training manual, but outside of MATE workshops, there is no support for additional MATE messaging. Taking a more holistic view of MATE, its delivery and potential impacts in a church setting offers opportunities to overcome the identified barriers. For example, social marketing campaigns have been used to successfully encourage prosocial bystander behavior [35,36] and may offer a way to reach community members who do not attend workshops. A well-formulated campaign could add structure and guidance to support for trainers and church leaders in the delivery of MATE content. In concert with delivery of MATE workshops (that have an appropriate religious context and have been adapted appropriately for the church’s needs), this multi-layered approach could be an important tool to raise awareness, promote gender equality, help with workshop recruitment and encourage prosocial bystander behavior change.

### 4.5. Undertake Co-Design

Successful implementation of a new intervention in an organization—in this case, including the MATE workshop in individual church ministries—requires an understanding of the behaviors and practices that need to change, the systems that are needed to support the implementation of the intervention, and the resources that are needed to facilitate this [37]. Gaining this understanding is more readily achieved by working with community members. Not only does this provide a level of insight unavailable to people outside the community, but it also ensures that the sociocultural needs and norms of the community are aligned with the program [38]. Additionally, this level of involvement will give the community more ownership of the program and contribute to the increased likelihood that it will be sustained [39]. Community involvement will be essential in any efforts MATE makes to provide religious context, support MATE adaptation in church communities, build leadership capability, and in church community information campaigns to identify the elements of the community that MATE is not familiar with and collaborate with MATE to find ways to address and accommodate those elements.

### 4.6. Actively Administer, Measure and Monitor

While the NFP partner coordinates church involvement in the MATE program, and MATE delivers the train-the-trainer workshop and provides ongoing engagement through the online communities, there is no central point actively administering the faith community program. Similarly, there is no mechanism for monitoring and evaluating what is delivered in each church, and so no way of implementing a system of continuous improvement in the program. Ongoing feedback about fidelity can help maintain community delivered programs that are closer to intended guidelines [28], so MATE would do well to develop a process for collecting information from trainers about changes that they make. Consideration of how MATE is being fit into church communities can inform the development of innovative and effective tools, guides and resources.

This research demonstrated that to realize the full potential of a model of MATE delivered through faith communities, there is a need for a program coordinator role with responsibility for the management of the pilot and ongoing support and engagement with the trainers and their churches.

### 4.7. Limitations and Future Research

Only 10 of the 31 trainers (29%) in the pilot were willing to participate in this study, so the experience and delivery of MATE for more than two-thirds of the church-based trainers in the pilot remains unknown. Additionally, volunteer bias may factor in this study, as trainers who participated may have been more engaged in the pilot, and so may hold different perspectives and have had different program experiences when compared to those of the unsampled trainers.

A thorough theory-based evaluation to examine how the program mechanisms identified in this study work in a faith context would provide valuable insight into the delivery of MATE in a faith setting and inform the scaling of this program to more churches and faith communities. Such an evaluation would require more active program administration to collect the necessary process information and more engagement from church-based trainers. Perspectives of MATE facilitators who deliver the program in secular and church settings would be valuable additions to this work.

## 5. Conclusions

In a climate where violence against women continues to be a significant social and public health problem, and where resources to fund prevention initiatives are scant, delivery of MATE through church communities offers a very practical opportunity. They possess resources, networks and knowledge that are essential for developing and delivering relevant and engaging public health interventions. Lessons learned in this pilot on the Gold Coast have highlighted that to capitalize further on this opportunity there will need to be some initial investment in tailoring the program for faith communities and building social campaign resources around MATE, as well as ongoing resourcing for a coordinator who can encourage and support church communities to implement MATE in ways that are meaningful, measurable and sustainable. These lessons can also inform other programs wishing to leverage faith communities, particularly regarding the importance of co-design in developing successful, sustainable interventions.

## Figures and Tables

**Table 1 ijerph-19-15833-t001:** Affiliations of Christian churches participating in the MATE pilot.

Affiliation	Number of Churches	
	In the Pilot	In This Study
Baptist	1	0
Church of Christ	1	1
Pentecostal	2	0
Seventh Day Adventist	2	2
Independent or no stated affiliation	6	4
Total	12	7

**Table 2 ijerph-19-15833-t002:** Characteristics of the study sample of church-based trainers (n = 10) in the Gold Coast MATE pilot.

**Age (Years)**	Mean	43.8
	Range	24–57
**Gender**	Male	5 (50%)
	Female	5 (50%)
**Average Weekly Household Income**	Less than $1000	2 (20%)
	$1000–$1500	1 (10%)
	$1500–$3000	7 (70%)
**Highest Level of Education**	TAFE/qualified trade	3 (30%)
	Bachelor’s degree	3 (30%)
	Post-graduate qualification ^1^	4 (40%)
**Cultural Heritage**	Aboriginal or Torres Strait Islander	
	Yes	0
	No	10 (100%)
	Cultural background	
	Australian	7 (70%)
	English	1 (10%)
	New Zealander	1 (10%)
	Scottish	1 (10%)

^1^ Graduate certificate, graduate diploma or master’s degree. Does not include a higher research degree (research master’s or doctorate).

**Table 3 ijerph-19-15833-t003:** Actual and planned MATE program delivery in participating churches.

Church	Status	Format	Content	Audience	No. of Trainers
C01	Delivered (×2)	Four weekly 2-h sessions delivered over consecutive weeks; standalone	*“[We] did the introduction… gender roles… [then] the Duluth wheel and all that sort of stuff on the second week. And then the third week there was a couple of, like, the inequality in sport, jokes and languages, just stuff like that. And then we tied it all up at the end.”*	Congregation –general	Three
C02	Delivered (×1)	One 3-h session; standalone	Not specified	Congregation –general	Two
C03	Delivered (×1)	One 2-h session; at regular church lunch	*“We did leadership, in the foundational exercises. We walked through what is domestic violence, bystander intervention. And then we did a group activity as well… it might have been abusive relationships that we did. We walked through the wheel of violence as well… that took us two hours...”*	Congregation –general	Two
C04	Delivered (×3)	One 3–4 h session; standalone	*“Along the lines of the foundational stuff that we did in the first half with, you know, applying some scenarios and then getting into the extracurricular, you know, I think we chose three through to do in the second half.”*	Church leadership/staff	Two/three
	Planned	One 3–4 h session; home group bible study	Not specified	Congregation –general	Two
C05	Delivered (×2)	Not specified	Not specified	Church leadership/staff	Two
	Planned	One 1-h session; part of a pre-existing regular youth evening	*“[The session] will go through everything from start to finish”*	Congregation –youth	Not specified
	Planned	One 4-h session	*“We’ll probably stick with the basics initially, with the youth side of things, to get them to understand what is reality versus what they’re seeing and looking at on social media, and in videos and movies… It will be really about consent, what consent means, the pornography side of things, along with advertising… one or two topics, and then next time a couple more topics”*	Church staff	Not specified
C06	Delivered (×1)	One 2-h session; part of a pre-existing men’s group	*“We did the introduction bit about what MATE was and what a bystander is all about, that stuff. We also did types–what is domestic violence and types of abuse. We did some brainstorming and some activities around that, and then they had a little break and talked about that a bit. And then they came back, and we did the gender box roles.”*	Congregation –men	Two
C07	Planned	One 3-h session; standalone	Not specified	Church staff	Two

**Table 4 ijerph-19-15833-t004:** Influences on the delivery of MATE workshops aligned to COM-B model components.

COM-B Component	Theme	Influence
CAPABILITY		
Psychological	Knowledge about violence against women	Understanding nature and drivers of violence against women
		Knowing bystander intervention techniques and their use
	MATE program knowledge	Knowing about MATE program structure and delivery
Physical	Facilitation skills	Having the skills to facilitate a MATE workshop
OPPORTUNITY		
Physical	Church environment	Having leadership support
		Acknowledgement of violence against women in community
		Church not ‘walking the talk’
		Familiarity and/or personal relationships with community
		Space in the church calendar
		Concerns about the nature of MATE program content
	Resourcing and logistics	COVID impacts
		Trainer time commitment
		Human resources to deliver training
Social	Christian context	Bible interpretation
		Church/secular divide
		Diversity within church community
	Gender and power dynamics	Patriarchal traditions
		Hierarchies within church
		Conservative views
	Support and encouragement	Church support
		NFP partner support
		MATE program support
		Peer (trainer) support
MOTIVATION		
Reflective	Christian identity	
	Responsibility and leadership	
	Making a difference	
Automatic	Passion	

## Data Availability

The data presented in this study are available on request from the corresponding author.

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
