# Peer review of "Leveraging Faith Communities to Prevent Violence against Women: Lessons from the Implementation and Delivery of the Motivating Action through Empowerment (MATE) Program"

_ijerph, 2022, doi:10.3390/ijerph192315833_

Round 1

Reviewer 1 Report

Data was well analyzed and synthesized and valuable recommendations were suggested after discussion of findings.

But I am little concerned whether the sample is sufficient for this type of research? It would be appropriate to mention the limitations of the research.

Few suggestions:

Line 101. Double "that". One can be deleted

Kindly provide some background info about MVP and MATE (e.g. when were they founded and by whom, who runs them, where do they operate, very brief background of MATE in Australia and Queensland, etc).

Was any response collected from resource persons delivering the workshop? e.g. Did they assess the context before preparing material? How different was their experience in this novel context from the others? etc,). If not, please mention why not.

Line 528. "to" may be deleted.

Author Response

Point 1: Data was well analyzed and synthesized and valuable recommendations were suggested after discussion of findings.

Response 1: Thank you.

Point 2: But I am little concerned whether the sample is sufficient for this type of research? It would be appropriate to mention the limitations of the research.

Response 2: Limitations have been added in at the end of the discussion (lines 534-540)

Point 3: Line 101. Double "that". One can be deleted

Response 3: Deleted

Point 4: Kindly provide some background info about MVP and MATE (e.g. when were they founded and by whom, who runs them, where do they operate, very brief background of MATE in Australia and Queensland, etc).

Response 4: The MVP description has been expanded (lines 48-53). Additional MATE background (lines 78-80) and delivery information (lines 87-88) has also been provided.

Point 5: Was any response collected from resource persons delivering the workshop? e.g. Did they assess the context before preparing material? How different was their experience in this novel context from the others? etc,). If not, please mention why not.

Response 5: The material used in the pilot was not modified for the church context. Text has been added (lines 119-121) to make this explicit.

Comparing the facilitators experiences delivering MATE in the church setting to secular settings they have delivered in would have provided an additional perspective for consideration in the development of the program for the church context. This was not considered in this study but has been noted as an area of future research (lines 546-547).

Point 6: Line 528. "to" may be deleted.

Response 6: Deleted.

Reviewer 2 Report

Very good article that covers an important topic from different angles. The existence of few studies on Violence prevention initiatives in faith settings make this article even more original. The framework, participants, background information, which and how methods to be used, and the pros and cons of the study are discussed in detail. Motivation influences, training delivery, improvement in the training program, relations with church leadership, monitoring the success of the program are also very well analysed. Quotations showing the negative and positive views of the participants about the program are also appropriate. In my opinion, the only point in the article that needs to be explained a little more is the views and comments on the Christian and non-Christian distinction in the MATE program. I would also expect the concluding remarks of this important article to be a little more inclusive and striking. Finally, ‘key words’ look like a sentence therefore they should be shortened, for example violence, women, etc instead of writing violence against women.

Author Response

Point 1: Very good article that covers an important topic from different angles. The existence of few studies on Violence prevention initiatives in faith settings make this article even more original. The framework, participants, background information, which and how methods to be used, and the pros and cons of the study are discussed in detail. Motivation influences, training delivery, improvement in the training program, relations with church leadership, monitoring the success of the program are also very well analysed. Quotations showing the negative and positive views of the participants about the program are also appropriate.

Response 1: Thank you.

Point 2: In my opinion, the only point in the article that needs to be explained a little more is the views and comments on the Christian and non-Christian distinction in the MATE program.

Response 2: Additional explanation added in (lines 351-352).

Point 3: I would also expect the concluding remarks of this important article to be a little more inclusive and striking.

Response 3: Thank you for this feedback. We have made some additions to the conclusion (lines 551-553 and 559-560) that emphasise the opportunity in faith communities and the importance of co-design for intervention success.

Point 4: Finally, ‘key words’ look like a sentence therefore they should be shortened, for example violence, women, etc instead of writing violence against women.

Response 4: Thank you for this feedback. We were guided by the existing literature in the selection of these keywords and believe that they offer an appropriate level of specificity for this paper, so have opted to leave them as originally submitted.